# Prostate cancer androgen receptor splice variant 7 biomarker study - a multicentre randomised feasibility trial of biomarker-guided personalised treatment in patients with advanced prostate cancer (the VARIANT trial) study protocol

Emma Clark [1], Miranda Morton,[2] Shriya Sharma,[2] Holly Fisher,[3] Denise Howel,[3] Jenn Walker,[2] Ruth Wood,[2] Helen Hancock,[2] Rebecca Maier,[2] John Marshall,[4] Amit Bahl,[5] Simon Crabb,[6] Suneil Jain,[7] Ian Pedley,[8] Rob Jones,[9] John Staffurth [10,11] Rakesh Heer[1]

JS and RH are joint senior authors.

For numbered affiliations see end of article.

**Correspondence to**
Dr Emma Clark;
emma.clark@ncl.ac.uk

## ABSTRACT

**Introduction** Prostate cancer is the most common male cancer with one in four developing non-curable metastatic disease. Initial treatment responses to hormonal therapies are transient and further management options lie between (1) further hormone therapy or (2) a non-hormonal approach involving additional chemotherapy or molecular radiotherapy (radium-223). There is no clear rationale for choosing between these mechanistically different treatment approaches. The biology of hormone resistance is driven through abnormal androgen receptor activity and we can assay this through a blood test measuring androgen receptor variant 7 (AR-V7) expression in circulating tumour cells. Despite increasing evidence supporting AR-V7's role as a prognostic marker, the clinical utility of such measures remains unknown in helping personalise treatment decisions.

**Methods and design** The VARIANT feasibility trial is a pragmatic design, to be run over 18 months with participants randomised into the intervention arm receiving biomarker (AR-V7) guided clinical treatment and participants randomised into the control arm with conventional standard management (no biomarker guidance). AR-V7 positive participants (likely to be insensitive to further hormone treatment) will receive chemotherapy or in other cases radium-223 (where routinely available). Seventy male ≥18 years old participants with metastatic castrate resistant prostate cancer clinically indicated to proceed to further hormone therapy or chemotherapy, will be recruited from three National Health Service Trusts based in England, Scotland and Wales. The feasibility primary outcome is willingness of patients to be randomised and clinicians to recruit to a biomarker-based treatment strategy, with trial data informing the basis of a definitive and appropriately powered randomised control trial.

**Strengths and limitations of this study**

► Focuses on a priority area of need in advanced prostate cancer clinical practice.
► To date, the feasibility of delivering a randomised biomarker guided-treatment trial in prostate cancer to formally assess clinical utility is not established and will be addressed in this study.
► As a feasibility study, the planned sample size (70 participants) does not have sufficient power or precision to compare the 'event' rate between treatment arms, but will allow informed planning for a definitive randomised controlled trial with prominent clinicians from non-recruiting centres involved in feasibility, to aid with follow-on trial.
► Emerging evidence points to additional androgen receptor biology driving hormone resistance, such as other variant expression and mutations — these along with alternative biomarkers can be explored in the associated biobanked samples (including cell-free tumour DNA).
► Strong patient and public involvement (PPI) to inform study design with a clear commitment to informing participants of project outcomes, setting a clear new gold standard for PPI.

**Ethics and dissemination** Formal ethics review was undertaken with a favourable opinion, through Wales NRES Committee 2 18/WA/0419. Findings to be disseminated through patient and professional organisations that have expressed their support, media outlets and peer-reviewed journal publication.

**Trial registration number** ISRCTN10246848; pre-results

## INTRODUCTION
### Background

Prostate cancer (PC) is the most common male cancer in the UK and the second highest cause of male cancer death.[1] In large part, PC is a slowly progressive disease and when detected at an early stage is managed by active surveillance, surgery or radiotherapy. However, 25% of patients will present with, or will progress to, advanced metastatic PC.[2 3] Metastatic PC is incurable, with less than one-third of the patients surviving more than 5 years.[1]

Medical castration (commonly referred to as hormonal treatment or androgen deprivation therapy (ADT)), blocks production of the hormone testosterone and/or targets the androgen receptor (AR) signalling axis that drives cancer cell growth. Although a good response to hormonal treatment seen often initially, disease progression to a lethal metastatic castration-resistant prostate cancer (mCRPC) is common.[4] Clinical trials have shown that the addition of chemotherapy (docetaxel) or other hormonal approaches (abiraterone acetate or enzalutamide) to initial hormonal therapy have led to a substantial improvement (ie, delay) in time to the development of mCRPC and overall survival (OS).[5–9] Furthermore, promising recent evidence from randomised trials of androgen-receptor axis-targeted drugs (ARATs) have shown addition of apalutamide (an inhibitor of the ligand-binding domain of the AR), alongside hormone therapy results in longer overall survival and radiographic progression free survival compared with placebo.[10] However despite these rapid advances, mCRPC typically manifests within 3 years and is uniformly fatal.[11–13]

### Treatment management for mCRPC

Management pathways for mCRPC are still evolving in response to emerging new treatments; however, it broadly follows one of two standard care approaches;[14] (1) further hormonal treatment such as abiraterone or enzalutamide or (2) 'non-hormonal' treatment, typically chemotherapy or molecular radiotherapy (radium 223) (where available). There is no clear biological rationale for choosing between these mechanistically different treatment approaches. Suitable patients for this study can receive both approaches in a sequential manner if one is failing. Patients and clinicians often prefer hormonal treatment, being less toxic and easier to manage, however, only 30% to 50% of men respond well, with the remainder demonstrating a poor or an equivocal response.[15 16] As many patients will not respond to either treatment approach, there are considerable costs from our current management pathways, both in terms of patient experience and outcomes (side effects and disease progression) and economic costs to the National Health Service (NHS) (large burden of expensive treatments for the the most common male cancer). Personalised management pathways are urgently needed.

### Biology of the androgen receptor (selective treatment pressure)

A breakthrough in understanding the biology of PC revealed that hormonal treatments generate a selective pressure at the cellular level inducing complex molecular mechanisms characterised by an adaptation of the AR signalling axis. This results in tumour resistance mediated by the induced expression of alternative types of androgen receptor. These AR messenger RNA (mRNA) splice variants lack the important hormone-binding domain, resulting in a constitutively active cellular receptor, despite castration. The most widely studied variant is androgen receptor variant 7 (AR-V7).[17 18] AR-V7 activity is not affected by 'hormonal treatment' such as enzalutamide and abiraterone that target the hormone-binding domain, potentially rendering these treatments ineffective in men with AR-V7.[19–21] A surge in ARATs available for clinical use (eg, apalutamide and darolutamide) will most likely enhance this burden (although of note, evidence demonstrating reduced effectiveness of these specific treatment in men who are positive for AR-V7 or other variant splice forms including AR point mutations, have not been published to date).

### Rationale

Published clinical data demonstrates a strong link between AR-V7 expression and mCRPC progression and highlights the potential for AR-V7 to be utilised as a treatment stratification biomarker to identify those men likely to be sensitive to further hormonal treatment (AR-V7 negative patients) and avoid futile treatments in those predicted to be insensitive (AR-V7 positive patients).[21–29] Notably AR-V7 positivity is not associated with insensitivity to taxane chemotherapy treatment (relative reduction in risk of death of 76% maintained with chemotherapy, HR: 0.24; 95% CI 0.10 to 0.57; p=0.035)[27 30] and data from the recent PROPHECY trial (Multicenter Prospective Trial of Circulating Tumor Cell AR-V7 Detection in Men with mCRPC Receiving Abiraterone or Enzalutamide, NCT02269982), reports on the prognostic value of the AR-V7 biomarker (prospective observational cohort of poor prognosis patients with advanced prostate cancer who receive abiraterone or enzalutamide treatment).[30] The commercially available AdnaTest ProstateCancerPanel AR-V7 assay (Qiagen) detects AR-V7 mRNA expression in circulating tumour cells (CTCs) in whole blood and has been independently and robustly clinically validated in terms of reproducibility and comparisons of sensitivity and specificity with other AR-V7 detection platforms.[31–33] However to date, there have been no formal measures of the clinical utility of AR-V7 as a predictive biomarker.

### Evidence gap

Encouragingly, a cost saving analysis of performing AdnaTest ProstateCancerPanel AR-V7 biomarker testing in mCRPC demonstrated use of the biomarker would result in a substantial cost saving as long as the true prevalence of AR-V7 was >5% (well below the accepted prevalence rate of

30%).[34] However, formal cost-effectiveness analyses based on incremental cost-effectiveness ratio (cost per quality-adjusted life year gained) and assessing prevalence rates of this biomarker have yet to be carried out.

The National Comprehensive Cancer Network Task Force[35] and Cancer Research UK (CRUK) consensus statement on biomarker roadmap for cancer studies,[36] have highlighted the key recommendations for accelerating a tumour biomarker into clinical practice by sequentially demonstrating evidence for; [1] analytical reproducibility; [2] clinical validity and [3] clinical utility. Previous clinical studies on AR-V7 testing focused on retrospective or prospective cohort analyses of associated AR-V7 expression distinguishing subgroups with different clinical outcomes with hormonal treatment in men with metastatic PC.[37–40] However, the highest level of assessment of clinical worth in improving patient outcomes (clinical utility) remains lacking. We have paid particular focus to address clinical utility evidence gaps in the VARIANT trial using published levels of evidence standards for assessing biomarkers to inform study design.[41 42] We aim to demonstrate improvement in patient outcome sufficiently to justify AR-V7 biomarker incorporation into routine clinical care (including feasibility of collecting quality of life measures for a future health economic evaluation).

### Emerging treatment landscape

The treatment landscape of hormone sensitive (PC) is evolving, altering treatment pathways for mCRPC. Recent data from the USA based PROPHECY trial reporting on a prospective observational cohort showed mRNA AR-V7 (modified Qiagen AdnaTest ProstateCancerPanel, Baltimore, Maryland) and protein AR-V7 (Epic nuclear-specific, San Diego) biomarker positivity associated with worse progression free survival and OS in poor prognosis patients with advanced prostate cancer who receive abiraterone or enzalutamide treatment.[30] Criticisms of the study included lack of testing with alternative treatment such as chemotherapy (which we have addressed in this study) and preselection of high risk CRPC patients (ie, those with poor prognosis), ultimately generating results that cannot be extrapolated over the overall CRPC population.[43–46] Of note, lower AR-V7 prevalence was reported in the overall CRPC population in the ARMOR3-SV phase III clinical trial which employed the AdnaTest Prostate-CancerSelect and detect CTC assay (Qiagen) to assess AR-V7 mRNA expression, where only 8% of men were AR-V7 positive (95% CI 6 to 10).[47 48] During reviewing of this protocol, results of the CARD trial (Cabazitaxel vs Abiraterone or Enzalutamide in Metastatic Prostate Cancer) were published showing median overall survival was 13.6 months with cabazitaxel and 11.0 months with androgen signalling targeted inhibitors (HR for death, 0.64; 95% CI 0.46 to 0.89; p=0.008). CARD investigators plan to analyse CTCs for AR-V7 in order to determine the prognostic and predictive value of CTC-derived AR-V7

detection, further contributing important findings from this evolving treatment landscape.[49]

We argue irrespective of the evolving treatment landscape, the opportunity to generate feasibility data for a biological (biomarker) informed approach to treatment selection over standard care protocol-based approaches, tests a highly relevant clinical question in these high risk CRPC patients (ie, those who have more to loose from pursuing a 'try and see' approach). This would provide an appealing long-term strategy (for patients and service providers) to ultimately improve on clinical outcome (specifically for a clinical subgroup of poor prognosis patients, identifying those likely to be sensitive to further hormonal treatment and avoid futile treatments in those that are predicted to be insensitive).

### Main aim of study

To determine the feasibility of conducting a definitive randomised control trial to evaluate the clinical utility of an AR-V7 blood biomarker assay in personalising treatment for men with mCRPC in UK NHS clinical practice.

### Objectives

Feasibility study.

#### Primary objective

1. To establish if it is feasible to conduct a definitive trial comparing AR-V7 biomarker-driven management with the current standard care in patients with mCRPC.

#### Secondary objectives

2. To estimate AR-V7 biomarker prevalence in the trial population to inform sample size calculations for a definitive randomised control trial.
3. To assess recruitment, compliance and retention rates.
4. To confirm outcome measures for a future definitive trial and establish trial data response rates, variability and data quality.
5. To establish a blood sample biorepository to include baseline, 12 and 24 week blood samples for future translational studies.

#### Exploratory objectives

6. To establish a complete serial blood tissue archive to include potential measures of cell-free DNA (cfDNA) and additional AR variants in CTCs and cfDNA biomarker measures to complement AR-V7 reads (such as AR mutations, other AR splice forms and AR amplification, and other mutations such as PTEN,p53,MYC gain,RB1 loss,MET gain and further molecular pathways yet to be defined), depending on the ultimate biomarker performance characteristic established in this trial population. Blood will be collected, processed and archived at 0 weeks (baseline), 12 weeks and 24 weeks following the first treatment.
7. To explore thresholds of the magnitude of AR-V7 positivity to investigate relationships with outcomes and estimate AR-V7 positivity rate assumptions regarding a cut-off point.

8. To undertake cross-site validation of biomarker reads between two GCP laboratories (Newcastle University and Cardiff University).

## METHODS AND ANALYSIS
### Study design
This feasibility study is a multicentre, two-arm, randomised control trial (RCT). All patients who consent to take part in the trial and who are eligible, will have a blood test to assess prevalence rate of the AR-V7 biomarker. Participants will be randomised in the ratio 1:1 to receive personalised standard treatment (intervention) guided by AR-V7 biomarker status or standard care (control) without biomarker-guided treatment. Those in the control group will not receive blood biomarker test results.

The treatment for each patient is expected to be dependent on various factors (eg, clinician choice, patient choice, previous treatments, comorbidities, concomitant medication and pattern of disease), as well as randomised allocation and AR-V7 status in the personalised treatment arm. All treatments are part of standard care for these participants. Treatment options for participants randomised to the personalised standard treatment arm will be recommended, but not mandated within this feasibility trial, with reasons for not following the recommendation recorded and reported as outcome. A

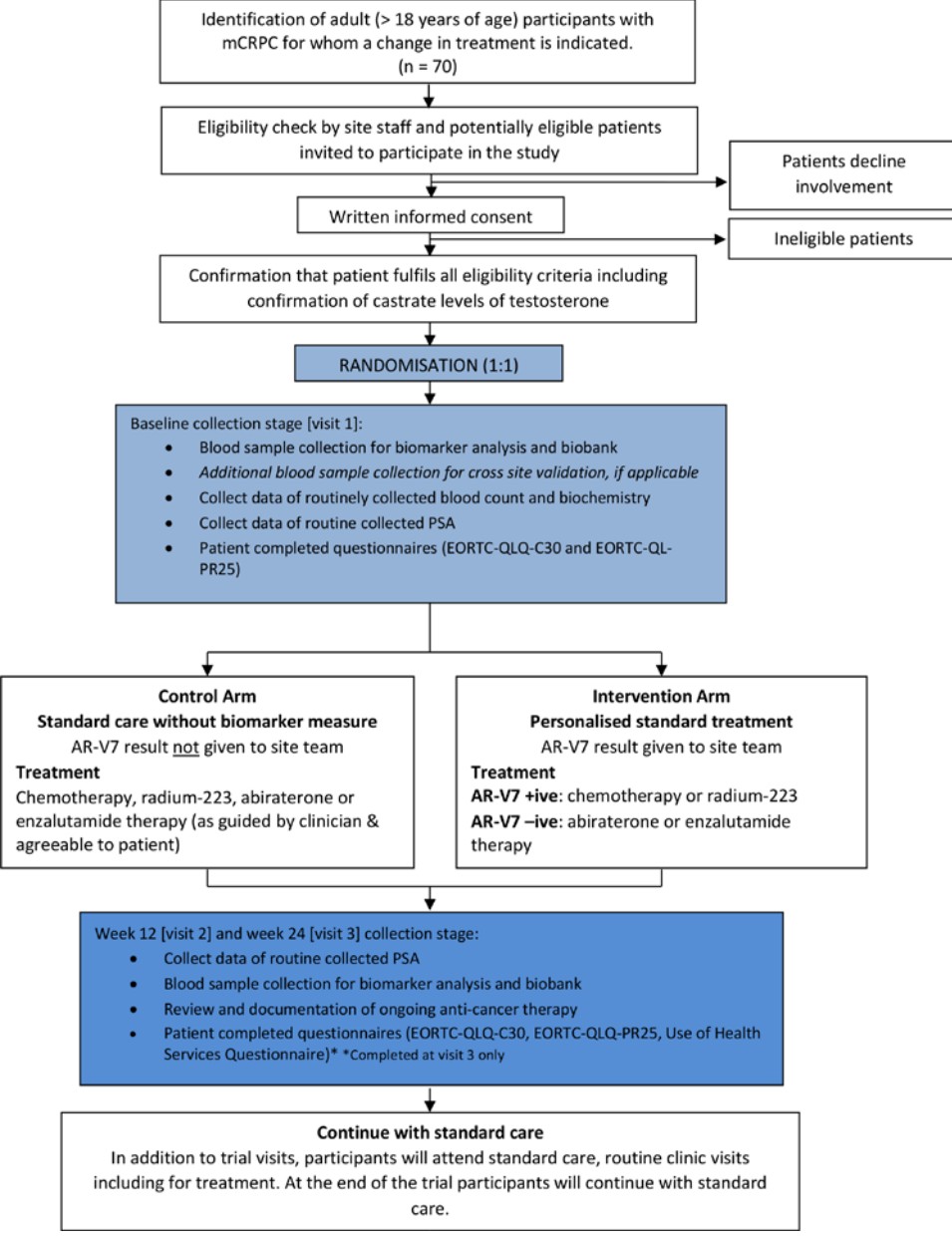

**Figure 1** VARIANT trial Consolidated Standards of Reporting Trials diagram. Planned flow of participants throughout the VARIANT study. AR-V7, androgen receptor variant 7; EORTC-QLQ-C30/PR25, European Organisation for Research and Treatment of Cancer Quality of Life Questionnaire; mCRPC, metastatic castrate resistant prostate cancer; PSA, prostate specific antigen.

Consolidated Standards of Reporting Trials diagram of study protocol (V.2.0, 8 March 2019) is shown in figure 1.

## Study setting

Seventy patients with mCRPC who require a change in treatment will be recruited in three secondary care NHS Trusts in the UK spread across England (The Newcastle upon Tyne Hospitals NHS Foundation Trust), Scotland (NHS Greater Glasgow and Clyde) and Wales (Velindre University NHS Trust). We aim to recruit mCRPC patients with a predicted poor overall survival. We anticipate this group of mCRPC patients have the most to gain from a biological-based treatment approach as their disease is more likely to progress during a period of treatment with an inactive agent. Multivariate analysis from the metastatic population of STAMPEDE[50] (Systemic Therapy in Advancing or Metastatic Prostate Cancer: Evaluation of Drug Efficacy a multi-arm multi-stage randomised controlled trial, NCT00268476), has shown that worse overall survival was seen in men with the following features: presence of bone metastases (regardless of soft tissue metastases), worse WHO performance status (0 vs 1 or 2), higher (or unknown) initial Gleason sum score category (≥8 vs ≤7) and younger age at randomisation <60 years. Poorer failure free survival (but not overall survival was additionally seen in men with worse primary tumour stage and higher PSA level before starting ADT. There is overlap between these poor prognostic features and factors associated with a high likelihood of harbouring AR-V7 positive CTCs.[33 40 51]

## Eligibility criteria

Patients will be aged ≥18 years old with metastatic castrate resistant prostate cancer (high risk features) clinically indicated to proceed to further hormone therapy or chemotherapy and fulfil all of the following criteria:

1. Histologically or cytologically proven diagnosis of adenocarcinoma of the prostate.
2. Radiographic and/or histological and/or cytological evidence of metastatic disease.
3. Castrate levels of testosterone and documented ongoing medical or surgical castration. Testosterone level ≤50 ng/dL/1.73 nmol/L and maintaining on androgen suppression therapy.
4. Disease progression since the last change in therapy defined by one or more of the following: (i) PSA progression as defined by the prostate cancer working group 3 criteria ≥2 ng/mL; (ii) bone disease progression as determined by the local radiology/multidisciplinary team; (iii) radiographic progression of nodal or visceral metastases as determined by the local radiology/multidisciplinary team.
5. Suitable for treatment with at least one novel hormonal treatment (with available treatments abiraterone acetate or enzalutamide) and one non-hormonal therapy (with available treatments docetaxel, cabazitaxel or radium-223).
6. At least two high risk features: (i) age <60 years at time of diagnosis of metastatic disease; (ii) bone metastases present at time of initial metastatic prostate cancer diagnosis (although not mandated, it is considered good clinical practice to have up to date imaging within 8 weeks); (iii) Gleason grade group 4 or 5 (Gleason score 8 to 10); (iv) presence of visceral metastases (eg, liver or lung) at any time point. This does not include lymph node metastases; (v) PSA doubling time <3 months; (vi) elevated alkaline phosphatase above institutional upper limit of normal; (vii) Eastern Cooperative Oncology Group (ECOG) Performance Status worse than or equal to 1; (viii) previous treatment for castration resistant prostate cancer with docetaxel chemotherapy; (ix) previous treatment for castration resistant prostate cancer with abiraterone and/or enzalutamide or equivalent agent.
7. Estimated life expectancy >6 months.
8. Provision of written informed consent, including consent for biobanking of blood samples.

## Exclusion criteria applied in the VARIANT trial are

1. Histological variants of prostate cancers with small cell or neuroendocrine features.
2. Prior or current malignancy (except adenocarcinoma of the prostate) with an estimated ≥30% chance of relapse/progression within next 2 years.
3. Previously identified brain metastases or spinal cord compression unless treated with full functional recovery.
4. Administration of an investigational agent within 30 days of first dose of trial medication.

## Randomisation

Patients will be randomised to receive either personalised standard treatment (guided by AR-V7 biomarker status) or standard care (not guided by biomarker status) on a 1:1 basis using a method of random permuted blocks of concealed variable block size and stratified by site.

## Study intervention

This three-centre randomised feasibility study incorporates a control and an intervention arm. All patients will undergo AR-V7 biomarker assessment with results only made known to the patients and clinical team in the intervention arm.

## Intervention arm

Treatment will be given as per standard care with recommendations guided by biomarker status; [1] if the participant is found to be AR-V7 positive, then non-hormonal treatment is recommended (docetaxel chemotherapy, cabazitaxel chemotherapy or radium-223 therapy) or; [2] if the participant is found to be AR-V7 negative, then next generation hormonal treatment is recommended (either enzalutamide or abiraterone).

The results of the AR-V7 biomarker assessment will be provided securely to the clinical team to enable tailored treatments based on AR-V7 expression from the biomarker

**Clinical outcome measures**

1. Time to PSA progression; Confirmed rising PSA more than 12 weeks after randomisation. (Where there has been a decline in PSA from baseline, progression will be a 25% or greater increase, and an absolute increase of at least 2 ng/mL, from the nadir, which is confirmed by a second value obtained 3 or more weeks later. Where no decline from baseline is documented, progression must be a 25% or greater increase from the baseline value along with an increase in absolute value of 2 ng/mL or more. In all cases, the initial rise in PSA must occur after a minimum of 12 weeks from randomisation).
2. Clinical progression and survival within 6 months; (i) Number of patients who have progressed clinically at 6 months (includes change of systemic anti-cancer therapy and death from prostate cancer); (ii) Cancer specific survival at 6 months and (iii) overall survival at 6 months.
3. Quality of life for patients with cancer (EORTC QLQ-C30 Questionnaire).
4. Additional quality of life items patients with prostate cancer (EORTC QLQ-PR25 Questionnaire).
5. Participant costs questionnaire (Use of Health Services Questionnaire).

result. Where a decision is made that the participant will receive a non-recommended therapy (either by the clinician or patient) this therapy, and the reasons for giving this, will be documented.

## Control arm

Participants with their clinical care team will make an informed and preference-based decision to receive standard care, including either next generation hormone treatments abiraterone or enzalutamide or non-hormonal approaches including docetaxel or cabazitaxel chemotherapy or radium-223. Details of all treatment administered, including doses, will be recorded as part of the trial.

The research team at sites will not receive the participants AR-V7 biomarker results.

## Outcome measures

Standardised clinical assessment tools used in monitoring CRPC disease and progression on treatment will be reported (listed in box 1). Primary outcome measures are related to feasibility (recruitment, retention and adherence) and will report the following;

1. The proportion of prostate cancer patients identified through clinics who meet the eligibility criteria.
2. The number of patients accrued per site per month over the course of the trial.
3. Baseline prevalence of AR-V7 expression in the participant cohort (this will be presented as a crude percentage of AR-V7 positivity of total participants, and in each arm).
4. The willingness of patients to be randomised (defined as the proportion of patients consenting to be randomised from all eligible patients approached about the study).

5. Compliance rate (this will be defined as the number of patients who start randomised treatment as a proportion of the number randomised).
6. The proportion of patients who: start AR-V7 recommended treatment; start treatment other than the recommended treatment; change treatment before disease progression or withdraw. (This measure will capture information regarding patients who choose not to take recommended treatment because of strong preferences and patients who progress rapidly while waiting for treatment with a change in eligibility for treatment options).
7. The proportion of trial participants with assessable blood samples for biomarker status (which would affect treatment targeting).
8. The median time from the blood sample being drawn to; (i) AR-V7 result being sent back to the site and (ii) patient starting treatment (and compared with standard of care treatment).
9. The proportion of randomised patients for whom data is collected on each clinical and health economic outcome at baseline, 12 and 24 weeks.

Further information on recruitment, screening, the patient consent procedure and informed consent literature, can be found in the online supplementary section.

## Data collection

Table 1 shows a trial schedule of events. A more detailed description of all data collection including a data management plan, can be found in the online supplementary section. In summary, in addition to collecting standard care assessment of disease status data from patients in the intervention and control arms, trial specific questionnaire assessment (EORTC-QLQ-C30 Quality of Life of Cancer Patients, EORTC-QLQ-PR25 Quality of Life Prostate Cancer Module), will take place at the baseline, 12 and 24 week visit.

## AR-V7 biomarker measure

A validated two-centre pipeline (consisting of preanalytical, analytical and postanalytical phases) to measure AR-V7 biomarker using the commercially available AdnaTest ProstateCancerPanel AR-V7 circulating tumour cell quantitative RT-PCR assay (Qiagen, intended for molecular biology applications), has been set up according to assay manufacturers recommendations, analytical methods and sponsor agreed Standard Operating Procedures (SOP's). Following biomarker data analysis and data verification, for participants randomised to the intervention standard treatment arm (AR-V7 biomarker guided), the baseline biomarker result and biomarker treatment recommendation will be sent securely within 10 working days to the local principal investigator and delegated research staff. Further information on the specifics of AR-V7 biomarker driven personalised treatment (sample receipt, processing, analysis and reporting of read-out), can be found in the online supplementary section.

**Table 1** Trial schedule of events

| Procedure | Screening | Visit 1 consent/ baseline | Visit 2 12 weeks (+/-2 weeks) | Visit 3 24 weeks (+/-2 weeks) |
|---|---|---|---|---|
| Medical history and demographics | X | | | |
| Record results of standard care PSA test | X | X | X | X |
| Eligibility assessment | X* | X* | | |
| Patient information sheet | X | | | |
| Informed consent | | X | | |
| Testosterone if no previous confirmation | | X† | | |
| Confirmation of eligibility | | X* | | |
| Randomisation | | X | | |
| Access to standard of care haemoglobin and biochemistry results | | X | | |
| Blood sample collection and shipment for CTC/ctDNA blood assessment and AR-V7 analysis (analysed at Newcastle University Central Analysis Lab) | | X | X | X |
| CTC blood sample collection and shipment for cross-site validation‡ (analysed at Cardiff University Central Analysis Lab) | | X‡ | | |
| EORTC QLQ-C30/PR25 Questionnaires | | X | | X |
| AR-V7 blood test result feedback to patient§ | | X§ | | |
| Use of Health Services Questionnaire | | | | X |
| Anti-cancer therapy review | | | X | X |
| Clinical assessment of disease status | | | X | X |

*Eligibility assessment performed against trial eligibility criteria in screening, patients likely to be eligible will be given a VARIANT information sheet and trial information. Eligibility will be confirmed by an Investigator (medically qualified doctor) after patients have provided written informed consent and before randomisation.

†In those cases where there is no previous confirmation of castrate levels of testosterone only. These patients will not be randomised until castration is confirmed and the patient is documented as eligible.

‡For selected patients only (confirmed at randomisation), for cross-site validation of AR-V7 status.

§For patients randomised to the personalised standard treatment arm (guided by AR-V7 biomarker) only.

AR-V7, androgen receptor variant 7; CTC, circulating tumour cells; ctDNA, circulating tumour DNA; EORTC-QLQ-C30/PR25, European Organisation for Research and Treatment of Cancer Quality of Life Questionnaire; PSA, prostate specific antigen.

## Data analysis plan

Analyses will be conducted on an intention-to-treat basis, with sensitivity analyses used to investigate the impact of non-compliance to allocated arm. Given the feasibility status of this study, all statistical analyses will be descriptive. The majority of the outcome data will be presented in simple descriptive tables presenting percentages, means and SD or five-number summary (as appropriate), for each arm of the study. Analysis of clinical and biomarker measures will be assessed by; [1] clinical progression and survival within 6 months; [2] PSA response/progression (confirmed rising PSA more than 12 weeks after randomisation); [3] clinical progression and survival (overall and cancer specific) within 6 months (includes change of cancer therapy for progression) and; [4] survival (overall and progression free) estimates will be derived using the Kaplan-Meier method and presented as 6 month rates with CIs. The relationship between survival estimates and continuous AR-V7 biomarker expression will be modelled considering non-linear transformations in a univariate Cox model, or parametric alternative, presented as parameter estimates (HR) with CIs.

Compliance with quality of life and health economic measures will be assessed by; [1] number of patients completing measures as a proportion of the number randomised and; [2] degree of completeness of each domain of the European Organisation for Research and Treatment of Cancer Quality of Life Questionnaire (EORTC-QLQ) and economic questionnaire measures. These scores will be presented graphically and with numeric descriptive statistics.

## Study statistical size calculations

This trial is designed as feasibility trial according to definition of Eldridge et al.[52] Feasibility includes the deliverability of the intervention and in this case, assessment of the frequency of the positive assay measurements (predicted at approximately 30%). It has been recommended that data in an external pilot trial is collected on a minimum of 60 patients per arm to estimate the 'event' rate.[53] However, we plan to calculate a pooled estimate of overall recruitment rate, and overall biomarker prevalence rate, and will recruit 70 patients in total to allow for attrition.

The performance of potential outcome measures for a definitive trial will be assessed by estimating data completeness of the instruments and any potential bias in the completion of follow-up data. This information will be used to inform the design, choice of outcomes, necessary sample size and approach to the analysis, of a future definitive trial.

## Safety reporting

This is a low risk trial and no specific safety reporting is required. Should an Investigator have any concern regarding participant safety as an outcome of their participation in the trial, they will contact the Trial Management Group (TMG) and Chief Investigator as soon as possible. The Trial Oversight Committee (TOC) will monitor concerns as required.

## Trial conduct and governance

The trial will be conducted in accordance with the UK Policy Framework for Health and Social Care and as applicable, the Guidelines for Good Clinical Practice. The TMG is responsible for the day-to-day management of the trial, overseeing all aspects of the conduct of trial to ensure that the protocol is adhered to and take appropriate actions to ensure patient and data safety. The TOC will review trial conduct and accumulating clinical trial data and provide overall supervision for the trial on behalf of the Sponsor and the Funder. The constitution

of the TMG and TOC including roles and responsibilities delegation for this trial can be found in the online supplementary section. Aggregated data will be analysed by the Trial Statisticians and reported to an external independent TOC at least annually.

## Public and patient involvement

The design, planning and management of this trial has been supported by two prostate cancer patient representatives (co-applicant on the funding grant and TMG member). Both have advocated the dissemination of trial findings to patients and ensured that the public was adequately considered during trial design. PPI has been embedded into the study, with the patient's voice a strong theme to inform and influence the ongoing research and development of participant information resources in collaboration with the 'Cancer Perspectives' patient representative group (Newcastle upon Tyne Hospitals NHS Foundation Trust). A strong commitment is to inform the participants of the outcome of this project, a clear new gold standard for PPI.

## Ethics and dissemination

All parties will conduct the trial in accordance with ethical opinion. No amendment to protocol will be made without consideration and approval by the Trial Management Committee.

Feasibility data will be published as a peer-reviewed article and if successful, these findings will contribute to gaining further funding for a HTA full trial. In addition, assessing clinical data and blood derivatives from the participant cohort will provide valuable material (circulating tumour cells transcript and plasma circulating tumour DNA), to validate translational studies of other AR aberrations and hormone targeting resistance pathways (or the emergence of biomarkers for chemosensitivity), to inform and contribute further to the rapidly evolving treatment developments for CRPC. Participants will remain anonymised in all publications.

We will also use dissemination through patient and professional organisations that have expressed their support for this trial (PCF, CRUK, NCRI Prostate CSG and BAUS) and through media outlets including web resources, lay press, academic national and international conferences and peer-reviewed journal publication.

**Author affiliations**
[1]Translational and Clinical Research Institute, NU Cancer, Newcastle University, Newcastle upon Tyne, UK
[2]Newcastle Clinical Trials Unit, Newcastle University, Newcastle upon Tyne, UK
[3]Population Health Sciences, Newcastle University, Newcastle upon Tyne, UK
[4]Trial Managment Group, VARIANT Trial, Newcastle-Upon-Tyne, UK
[5]University Hospitals Bristol NHS Foundation Trust, Bristol, UK
[6]University of Southampton, Southampton, UK
[7]Queen's University Belfast, Belfast, UK
[8]Newcastle Upon Tyne Hospitals NHS Foundation Trust, Newcastle Upon Tyne, UK
[9]University of Glasgow, Glasgow, UK
[10]Research, Velindre Cancer Centre, Cardiff, UK
[11]Division of Cancer and Genetics, Cardiff University School of Medicine, Cardiff, UK

**Acknowledgements** We would like to acknowledge the valuable support of the VARIANT Trial Oversight Committee (TOC) members Dr Alison Tree, Dr Andrew Feber, Dr Rhian Gabe and our patient representatives on this committee, Robin Millman and Paul Nash. We are grateful for the assistance and support of our scientific collaborators Scott Dehm (University of Minnesota), Luke Gaughan (NICR), Jun Luo (John Hopkins), Gert Attard (UCL) and the All Wales Medical Genetics Laboratory (AWMGL). Management of the study is by Newcastle University Clinical Trial Unit (NCTU).

**Contributors** EC: Conception and design of the work, data collection, drafting of the article, critical revision of the article and final approval of the version to be published. MM: Trial management, data collection and drafting of the article. SS: Trial management, data collection and drafting of the article. HF: Conception and design of the work, drafting of the article and critical revision of the article. DH: Data collection, drafting of the article, critical revision of the article and final approval of the version to be published. JW: Trial management, data collection and drafting of the article. RW: Trial management, drafting of the article, data collection, critical revision of the article and final approval of the version to be published. HH: Critical revision of the article and final approval of the version to be published. RM: Critical revision of the article and final approval of the version to be published. JM: Conception and design of the work, critical revision of the article and final approval of the version to be published. AB: Conception and design of the work. SC: Conception and design of the work and critical revision of the article. SJ: Conception and design of the work and critical revision of the article. IP: Conception and design of the work. RJ: Conception and design of the work, data collection, drafting of the article, critical revision of the article and final approval of the version to be published. JS: Conception and design of the work, data collection, drafting of the article, critical revision of the article and final approval of the version to be published. RH: Conception and design of the work, drafting of the article, critical revision of the article and final approval of the version to be published.

**Funding** This work is supported by a National Institute for Health Research (NIHR) Research for Patient Benefit (grant number PB-PG-0816-20043). The VARIANT trial is sponsored by Newcastle upon Tyne Hospitals National Health Service Foundation Trust.

**Competing interests** SJ reports personal fees from Astellas, personal fees from Bayer, personal fees from Janssen, personal fees from Boston Scientific, personal fees from Almac Diagnostics, personal fees from Sanofi Genzyme, personal fees from Movember, outside the submitted work. AB reports research funding and advisory roles with Sanofi and Janssen and an advisory role with Astellas and Bayer, outside the submitted work. RJ reports grants and personal fees from Astellas, grants and personal fees from AstraZeneca, personal fees and non-financial support from Bristol Myers Squibb, grants, personal fees and non-financial support from Bayer, grants and personal fees from Exelixis, personal fees and non-financial support from Janssen, personal fees and non-financial support from Ipsen, personal fees from Merck Serono, personal fees and non-financial support from MSD, personal fees from Novartis, personal fees from Pfizer, grants and personal fees from Roche, personal fees from Sanofi Genzyme, personal fees from EUSA, outside the submitted work. JS reports non-financial support from Bayer and personal fees from Janssen and Astellas outside of the submitted work. SJC has an honoraria/advisory role with Roche, Clovis Oncology, Bayer, Janssen Cilag and Merck and receives research support from AstraZeneca, Astex Pharmaceuticals and Clovis Oncology.

**Patient consent for publication** Not required.

**Ethics approval** Favourable ethical opinion has been obtained from the Wales National Research Ethics Service (NRES) Committee 2 18/WA/0419.

**Provenance and peer review** Not commissioned; externally peer reviewed.

**ORCID iDs**
Emma Clark http://orcid.org/0000-0003-0065-1463
John Staffurth http://orcid.org/0000-0002-7834-3172

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
