## [Reviewer comments · BMJ Open]

ARTICLE DETAILS

TITLE (PROVISIONAL)	The Prostate Cancer Androgen Receptor Splice Variant 7 Biomarker Study - A multicentre randomised feasibility trial of biomarker-guided personalised treatment in patients with advanced prostate cancer (The VARIANT Trial) study protocol.
AUTHORS	Clark, Emma; Morton, Miranda; Sharma, Shriya; Fisher, Holly; Howel, Denise; Walker, Jenn; Wood, Ruth; Hancock, Helen; Maier, Rebecca; Marshall, John; Bahl, Amit; Crabb, S; Jain, Suneil; Pedley, Ian; Jones, Rob; Staffurth, John; Heer, Rakesh

VERSION 1 – REVIEW

REVIEWER	Arun Azad Peter MacCallum Cancer Centre, Australia
REVIEW RETURNED	23-Oct-2019

GENERAL COMMENTS	This is an interesting study. I have a few specific questions: 1. In the ARMOR-3 trial, the PSA RR to Enza in AR-V7+ patients was 42%. Other studies have also raised some doubts about AR-V7+ patients being insensitive to Enza or Abi. In this context, how do the authors justify the current biomarker-directed treatment allocation plan?2. Why are the authors not collecting any translational blood tests on the control group? At the very least, determining AR-V7 status at baseline would be of interest (could be done retrospectively of course). This seems like a missed opportunity to obtain prospectively collected samples assessing AR-V7 status3. The authors state that the data from this trial will be will be "used to inform the design, choice of outcomes, necessary sample size and approach to the analysis, of a future definitive trial". However, it is not clear to me how the authors will determine if the results from this study justify moving forwards with a definitive trial. Can they please clarify?
---

REVIEWER	Kate Mahon Chris O'Brien Lifehouse, Australia
REVIEW RETURNED	30-Oct-2019

GENERAL COMMENTS	Commendable study addressing clinical utility of biomarker which is well described in the manuscript.
---

REVIEWER	Emmanuel S. Antonarakis, M.D. Johns Hopkins University, USA
REVIEW RETURNED	01-Nov-2019

GENERAL COMMENTS	This is an excellent study protocol which addresses an unmet medical need in the management of advanced prostate cancer: when to choose an AR-directed therapy and when to choose a chemotherapy (or other non-AR therapy) by using a blood-based biomarker (AR-V7) to help with this clinical decision. The only comment that I have is: In the Introduction or Discussion, the authors should mention the recent publication of the CARD trial (de Wit R, de Bono J, Sternberg CN, et al. NEJM 2019) that randomized patients to cabazitaxel vs AR-directed therapy. The reason that this trial is particularly relevant to this protocol is that the CARD investigators plan to analyze CTCs for AR-V7 in order to determine the prognostic and predictive value of CTC-derived AR-V7 detection in that context. Other than that, this is an excellent protocol and manuscript. I congratulate the study team on a job well done, and I wish them every success in completing their planned trial.
---

VERSION 1 – AUTHOR RESPONSE

Response to Reviewers:

Reviewer: 1

Reviewer Name: Arun Azad

Institution and Country: Peter MacCallum Cancer Centre, Australia Please state any competing interests or state 'None declared': Nil

Please leave your comments for the authors below:

This is an interesting study. I have a few specific questions:

1. In the ARMOR-3 trial, the PSA RR to Enza in AR-V7+ patients was 42%. Other studies have also raised some doubts about AR-V7+ patients being insensitive to Enza or Abi. In this context, how do the authors justify the current biomarker-directed treatment allocation plan?

Authors response to reviewer 1 (Q1):

Thank you reviewer 1 for highlighting the current uncertainty of AR-V7 use in patients, we agree the picture is complicated. A recent publication from Howard Scher's Memorial Sloan group confirms patients with detectable nuclear localised AR-V7 in CTCs had superior survival on next-generation AR inhibitors (*median 9.8 vs 5.7 mo: p=0.041*) Graf *et al.* Euro Urol (2019), proposing use of a AR-V7 CTC test to inform treatment choice has clinical value. We have added additional sentences in our protocol introduction updating our literature review to reflect the need for studies like the VARIANT trial, to explore definitively the clinical utility of using such biomarkers.

2. Why are the authors not collecting any translational blood tests on the control group? At the very least, determining AR-V7 status at baseline would be of interest (could be done retrospectively of course). This seems like a missed opportunity to obtain prospectively collected samples assessing AR-V7 status.

Authors response to reviewer 1 (Q2):

Absolutely we 100% agree this would be a missed opportunity and this work will be done as part of the translational study of the VARIANT trial. A prevalence rate of AR-V7 in both study

arms at baseline will be reported as a study outcome – see outcome (3) *baseline prevalence of AR-V7 expression in the participant cohort (this will be presented as a crude percentage of AR-V7 positivity of total participants, and in each arm)*.

3. The authors state that the data from this trial will be will be "used to inform the design, choice of outcomes, necessary sample size and approach to the analysis, of a future definitive trial". However, it is not clear to me how the authors will determine if the results from this study justify moving forwards with a definitive trial. Can they please clarify?

Authors response to reviewer 1 (Q3):

We will use biomarker acceptability to clinicians (proportion of prostate cancer patients identified through clinics who meet the eligibility criteria); biomarker acceptability to patients (proportion of patients consenting to be randomised from all eligible patients approached about the study); compliance rate and prevalence of biomarker in the cohort to inform power calculations for the formal follow on study (considering the implications of a biomarker strategy design, *Freidlin, JNCI 2010*).

Reviewer: 2

Reviewer Name: Kate Mahon

Institution and Country: Chris O'Brien Lifehouse, Australia Please state any competing interests or state 'None declared': None declared

Please leave your comments for the authors below:

Commendable study addressing clinical utility of biomarker which is well described in the manuscript.

Authors response to reviewer 2:

Thank you reviewer 2 for your supportive comments of this study.

Reviewer: 3

Reviewer Name: Emmanuel S. Antonarakis, M.D.

Institution and Country: Johns Hopkins University, USA Please state any competing interests or state 'None declared': None declared

Please leave your comments for the authors below :

This is an excellent study protocol which addresses an unmet medical need in the management of advanced prostate cancer: when to choose an AR-directed therapy and when to choose a chemotherapy (or other non-AR therapy) by using a blood-based biomarker (AR-V7) to help with this clinical decision. The only comment that I have is: In the Introduction or Discussion, the authors should mention the recent publication of the CARD trial (de Wit R, de Bono J, Sternberg CN, et al. NEJM 2019) that randomized patients to cabazitaxel vs AR-directed therapy. The reason that this trial is particularly relevant to this protocol is that the CARD investigators plan to analyze CTCs for AR-V7 in order to determine the prognostic and predictive value of CTC-derived AR-V7 detection in that context. Other than that, this is an excellent protocol and manuscript. I congratulate the study team on a job well done, and I wish them every success in completing their planned trial.

Authors response to reviewer 3:

Thank you reviewer 3 for bringing the CARD trial to our attention. This excellent trial was published whilst our manuscript was under review and we are delighted to now include this reference in the final draft of our protocol. We have inserted the following text between line 181 and 188 in the introduction section of the manuscript, and updated the reference list:

During reviewing of this protocol, results of the CARD trial (Cabazitaxel versus Abiraterone or Enzalutamide in Metastatic Prostate Cancer) were published showing median overall survival was 13.6 months with cabazitaxel and 11.0 months with androgen signalling targeted inhibitors (hazard ratio for death, 0.64; 95% CI, 0.46 to 0.89; P=0.008). CARD investigators plan to analyse CTCs for AR-V7 in order to determine the prognostic and predictive value of CTC-derived AR-V7 detection, further contributing important findings from this evolving treatment landscape(56).

VERSION 2 – REVIEW

REVIEWER	Arun Azad Peter MacCallum Cancer Centre
REVIEW RETURNED	22-Nov-2019
GENERAL COMMENTS	All comments addressed satisfactorily.